# Construction of Tongue Image-Based Machine Learning Model for Screening Patients with Gastric Precancerous Lesions

**DOI:** 10.3390/jpm13020271

**Published:** 2023-01-31

**Authors:** Changzheng Ma, Peng Zhang, Shiyu Du, Yan Li, Shao Li

**Affiliations:** 1Institute of TCM-X/MOE Key Laboratory of Bioinformatics, Bioinformatics Division, BNRist/Department of Automation, Tsinghua University, Beijing 100084, China; 2Department of Gastroenterology, China-Japan Friendship Hospital, Chaoyang District, Beijing 100029, China; 3Department of Traditional Chinese Medicine, Yijishan Hospital of Wannan Medical College, Wuhu 241000, China

**Keywords:** precancerous lesions of gastric cancer, tongue image, deep learning, disease screening, risk prediction, artificial intelligence

## Abstract

Screening patients with precancerous lesions of gastric cancer (PLGC) is important for gastric cancer prevention. The accuracy and convenience of PLGC screening could be improved with the use of machine learning methodologies to uncover and integrate valuable characteristics of noninvasive medical images related to PLGC. In this study, we therefore focused on tongue images and for the first time constructed a tongue image-based PLGC screening deep learning model (AITongue). The AITongue model uncovered potential associations between tongue image characteristics and PLGC, and integrated canonical risk factors, including age, sex, and Hp infection. Five-fold cross validation analysis on an independent cohort of 1995 patients revealed the AITongue model could screen PLGC individuals with an AUC of 0.75, 10.3% higher than that of the model with only including canonical risk factors. Of note, we investigated the value of the AITongue model in predicting PLGC risk by establishing a prospective PLGC follow-up cohort, reaching an AUC of 0.71. In addition, we developed a smartphone-based app screening system to enhance the application convenience of the AITongue model in the natural population from high-risk areas of gastric cancer in China. Collectively, our study has demonstrated the value of tongue image characteristics in PLGC screening and risk prediction.

## 1. Introduction

Gastric cancer is the second leading cause of cancer death in China, and more than 80% of patients are diagnosed at an advanced stage [1]. Patients with precancerous lesions of gastric cancer (PLGC), including intestinal metaplasia and dysplasia [2,3], suffer a higher risk of gastric tumorigenesis, with an annual incidence of 0.25–6% [4,5,6]. Screening and conducting reasonable health surveillance for patients with PLGC in the natural population would make great contribution to facilitating the early prevention of gastric cancer.

Current screening methods suffer from some challenges, including invasiveness and relatively low accuracy, which limits their applications in population screening. On the one hand, although gastroscopy and biopsy are the gold standards for gastric disease diagnosis [7], these methods remain inefficient and unfeasible for gastric disease screening [8]. As previous studies indicated, approximately half of the patients screened with gastroscopy are non-atrophic gastritis, and the early diagnosis rate of gastric cancer remains less than 20% [1,9]. On the other hand, the application of serum markers that are commonly used as screening factors in various gastric cancer risk assessment methods, such as pepsinogen I/II and gastrin-17 [10,11,12], has been limited for risk screening in natural populations due to the high sensitivity and specificity thresholds required [13]. In addition, it is not cost-effective to use either serum pepsinogen test screening or endoscopy as they difficulty in their practical application [14]. Screening high-risk groups for gastroscopy could triage patients and effectively improve the utilization efficiency of medical resources. Thus, considering the requirements of large-scale screening, screening methods with a high cost-effectiveness ratio and high accuracy are urgently needed to enhance their popularization [15].

As non-invasive indicators, tongue image characteristics have been used for the surveillance of a broad spectrum of diseases, inspired by the diagnosis experience in traditional Chinese medicine (TCM) [16,17,18,19,20]. Tongue image characteristics, including shape, color, and tongue coating, are believed to reflect the health condition, or the severity and progress of disease, especially for digestive diseases as the tongue is anatomically connected to the digestive system organs. For example, recent studies have indicated that tongue image characteristics show correlations with gastroscopic observations and could be used to predict gastric mucosal health [21,22]. In addition, it was revealed that tongue surface and color characteristics could be used as indicators to assist in gastric cancer diagnosis [23,24]. Moreover, morphological markers based on tongue images are considered to be valuable for risk screening for other diseases, such as diabetes, fatty liver disease, and COVID-19 [17,25,26,27]. From the pathologic and etiologic perspectives, the distribution of microorganisms on the tongue coating has also been found to be related to gastric diseases, which helps uncover non-invasive microbial markers for gastric disease risk screening [28,29,30]. The above studies demonstrate the great potential of tongue image characteristics in assisting disease screening. Therefore, uncovering the risk characteristics of tongue images is potentially valuable for constructing PLGC screening models.

Recently, deep learning techniques are widely used in building biomedical image-based disease screening and prediction models [31,32,33,34,35,36,37]. For example, some studies have applied deep learning to predict diverse cancer types, including prostate cancer and rectal cancer, based on medical images [33,38]. Using tongue images, some studies have applied deep learning techniques to identify risk features in tongue images for the detection of diseases such as stomach cancer and diabetes [23,39]. Therefore, deep learning techniques could be a pivotal tool to uncover the risk characteristics from tongue images, and further constructed a machine learning-based screening model.

Therefore, to improve the efficiency of screening patients with PLGC, particularly in natural populations, this study aimed to build a machine learning-based PLGC screening model which introduces tongue image information on the basis of existing risk indicators. In detail, we firstly explored the tongue image characteristics of patients with PLGC and integrated them with canonical screening indicators to develop a PLGC screening model called AITongue. We then evaluated its screening effect by external validation in an independent cohort and finally explored its potential value as a risk predictor of PLGC in a follow-up cohort. To our best knowledges, the AITongue model we have developed should be the first tongue image-based machine learning model for PLGC screening and risk prediction. We believe that our study will pave the way to addressing the urgent need for non-invasive PLGC screening in clinical practice.

## 2. Materials and Methods

### 2.1. Patient Enrollment, and Data Collection

Patients were enrolled in this study at the China-Japan Friendship Hospital and Yijishan Hospital of Wannan Medical College from 2015 to 2022. The experimental protocol was established according to the ethical guidelines of the “Declaration of Helsinki” and was approved by the Human Ethics Committee of the Institution Review Board of Tsinghua University (protocol code 20200069). Inclusion criteria: At least 18 years of age, clear language skills, no barriers in communication and willingness to accept clinical investigation and sign informed consent. Exclusion criteria: The presence of heart, cerebrovascular, liver, kidney, hematopoietic system diseases. 

### 2.2. Gastroscopy and Histological Examination

Using video endoscopes (Olympus Corp), upper gastroscopic examinations were performed by two gastroenterologists. Tissue samples for biopsy were reviewed blindly by the two pathologists according to the criteria proposed by the Updated Sydney System and the Chinese Association of Gastric Cancer [40,41]. The results of each biopsy were reported as normal, superficial gastritis, chronic atrophic gastritis, intestinal metaplasia, intraepithelial neoplasia, or gastric cancer, and each participant was assigned a global diagnosis based on the most severe gastric histologic finding among any biopsy. Helicobacter pylori (Hp) infection status was determined by enzyme-linked immunosorbent assay for plasma IgG [42].

### 2.3. Data Pre-Processing and Data Structuring

As the pivotal step for data pre-processing, a deep-learning model was constructed to identify and segment tongue bodies in raw images while excluding face and background information. Here, we trained the tongue body recognition and locating model using the YOLOv5 model and 180 tongue images that were labeled by TCM physicians with a square frame using “labelImg” software [43]. The YOLOv5 model is a common deep learning model for target detection which can accurately identify and locate the position of specific objects after training. Furthermore, using this model, we carried out tongue body recognition and cutting on tongue images, reshaped the images to 224 × 224, and formed a pre-processed tongue body image dataset. In this way, we could segment the tongue from the complex background to reduce the impact of the background on classification and improve accuracy. Python (3.7.0) and PyTorch were used for the tongue image preprocessing. Using this model, the tongue images were detected and cut into tongue body images for subsequent analysis.

Additional clinicopathological characteristics of the enrolled patients were obtained from electronic medical records. The obtained characteristics included basic information (gender, age) and symptom characteristics (xerostomia, bitter taste, gastric distention, stomach pain, etc.). All the above indicators were structured as two-category labeled data. Among them, age was divided into >50 and ≤50 years based on the median of age distribution. Multiple interpolation methods were used to fill in the missing data. Tongue labels (fissure, etc.) were assigned by physicians.

### 2.4. PLGC Screening Model Construction

The PLGC screening model was constructed following two main steps: image classification with a deep learning model, and data integration with a logistic regression model.

Firstly, the image classification model was constructed with the ResNet50 deep learning model [44]. The ResNet50 model has a wide range of applications and good performance in the field of image classification as it can introduce the residual blocks. In our study, the residual blocks of the ResNet50 model are structured as two bottlenecks (BTNK), designated as BTNK 1 and BTNK 2. Their structure diagram is shown in Appendix A, where CONV is the convolution block, BN is the batch normalization block, and Relu is an activation function in the bottleneck. After the ResNet50 module, tongue images were classified into two categories: high-risk and low-risk. 

A logistic regression model was then used to predict the PLGC screening results by integrating the tongue image classification results and the clinicopathological indicators. Logistic regression models have good performance in the integration of a small number of variables and robust prediction of classification tasks, resulting in their wide application in disease classification and risk prediction research. 

### 2.5. Statistical Analysis

All analysis procedures were performed using Python (3.7.0) and the sklearn package. Tongue diagnostic labels (TDL) and clinical symptoms with statistical significance (*p* < 0.05) by both univariate and multivariate analyses were included in the model. The significance of each factor adjusted for gender and age was calculated in the multivariate analysis. Binary logistic regression was used to construct the screening models. Chi-square tests were applied to calculate the significance of the independent variables for PLGC. Pearson’s correlation coefficient was applied to evaluate the correlation between the independent variables. Accuracy, sensitivity, specificity, recall, precision, receiver operating characteristic (ROC) curve, and area under the curve (AUC) were used as evaluation metrics to evaluate model performance.

AUC-ROC curves are performance measures for classification problems under various thresholds. ROC is a probability curve, and AUC represents the degree or measure of separability. The horizontal coordinate of the ROC curve is the false positive rate (FPR), and the vertical coordinate is the true positive rate (TPR). The calculation formula is as follows.
$$TPR=\frac{TP}{TP+FN}\;FPR=\frac{FP}{FP+TN}$$

TP, FP, TN, and FN represent true positives, false positives, true negatives, and false negatives, respectively. In the classification task, the model represents the prediction and ground truth. The higher the AUC, the better the classification performance of the model.

## 3. Results

### 3.1. The Overall Design of Our Study

In our study, a total of three cohorts of patients were enrolled with undergoing gastroscopy and pathology. These included a development cohort, validation cohort, and follow-up cohort. Here, two categories, including PLGC and non-PLGC, were derived for each patient based on pathological diagnosis (Table 1). In detail, we developed the PLGC screening model and performed an internal cross-validation on the development cohort, which consisted of 325 patients, including 55 PLGC and 270 non-PLGC patients. We then performed external validation on the validation cohort, which had a total of 1995 patients, including 171 PLGC and 1824 non-PLGC patients. It should be noted that we also evaluated the risk prediction value of the PLGC screening model on the follow-up cohort, in which only non-PLGC patients were enrolled at the baseline timepoint, and were further classified as Pro or non-Pro according to the pathological lesions at the endpoint after a mean follow-up time of 22 months (Figure 1). 

### 3.2. Construction of AITongue Model with Integrating Tongue Image Characteristics

After preprocessing the tongue images with deep learning (Figure 2a), a PLGC screening model, which we named the AITongue model, was constructed in the development cohort by integrating the tongue image and clinicopathological characteristics, as shown in Figure 2b. Here, the tongue images were classified into two categories: high-risk and low-risk. The AITongue model took the categorized results, and canonical gastric cancer risk indicators (age, gender, and Hp infection) as the input, and the PLGC prediction results as the output.

Firstly, we examined the screening value in terms of inclusion of tongue image characteristics in the development cohort. Through a five-fold cross-validation, it was shown that the AITongue model exhibited good performance in distinguishing PLGC from non-PLGC patients, with an accuracy of 0.69 and an AUC value of 0.75. In contrast, the screening model only including baseline indicators (age, sex, Hp) exhibited an accuracy of 0.60, with an AUC value of 0.69 (Figure 2c,d). Thus, it was revealed that the screening performance was improved by 8.7% with the inclusion of tongue image characteristics (Figure 2d), indicating the contribution of tongue image characteristics in PLGC screening.

We then further investigated and interpreted the tongue image characteristics with PLGC screening potential. Through performing correlation analysis between image risk classification (high vs. low-risk) obtained by the deep-learning model and TDL labels generated by TCM experts, we found that five of the TDLs were statistically significant (*p* < 0.05), namely greasy, fissured, dark, coating (yellow), and coating (thick). This indicated, to some extent, the medical significance of the risk features found in tongue images and suggests that there may also be some value of TDLs for PLGC screening (Table 2).

### 3.3. External Validation of PLGC Screening

We then validated the performance in PLGC screening of the AITongue model in the independent validation cohort. To enhance the robustness and application value of our model, the five representative TDLs that included greasy, fissured, dark, coating (yellow), and coating (thick), rather than the whole image characteristics, were selected as inputs for the AITongue model. Of note, it was found that these five TDLs, along with gender and age, showed significant correlations with PLGC in both univariate and multivariate analyses (Table 3, Appendix A), supporting their value as input parameters for AITongue.

It was found that the AITongue model showed a comparably discriminative performance in the independent validation cohort compared with that of the development cohort. Here, the AITongue model exhibited an accuracy of 0.64 and AUC of 0.75 in distinguishing PLGC from non-PLGC patients. In contrast, the model with only including baseline indicators (age, sex, Hp) showed an accuracy of 0.53 and AUC of 0.68 (Figure 3). Thus, we could conclude that the discriminative performance between PLGC and non-PLGC has been significantly enhanced by 10.3% (0.68 vs. 0.75, *p* < 0.01, Figure 3) by introducing tongue image characteristics. The results furtherly validated the effectiveness of tongue image characteristics for PLGC screening.

In addition, we also focused on clinical symptom characteristics and investigated their screening value for PLGC and confounding effects for the AITongue model [45]. 

First, we investigated the screening value of symptom characteristics by analyzing their associations with PLGC. As a result, three symptom characteristics (xerostomia, bitter taste, belching) showed significant correlations with PLGC in both univariate and multivariate analyses, whereas the others, including stomach pain, bloating, chilliness, and loose stools, did not show significant correlations (Appendix A). 

Further, we incorporated these three symptom characteristics into the AITongue model to evaluate the enhancement of introducing symptom characteristics for PLGC screening. The validation cohort was used as training data to construct a logistic regression model and a five-fold cross-validation was performed. It showed a small improvement in the discrimination between PLGC and non-PLGC after introducing symptom characteristics (0.76 vs. 0.73, Figure 4). These results indicate that the introduction of clinical symptom characteristics could improve the screening efficiency of PLGC, with a slightly lower effect than tongue image characteristics. In addition, there was a low correlation between tongue image and symptom characteristics (Appendix A), which indicates that tongue image characteristics might be independent factors from clinical symptom characteristics in terms of PLGC screening. 

### 3.4. Evaluation of the Validity of Tongue Image Characteristics for Risk Prediction of PLGC

PLGC risk prediction is pivotal for gastric cancer early prevention. Thus, we further explored the value of the AITongue model in predicting the risk of PLGC. Here, we enrolled a cohort of non-PLGC patients and conducted a long-term follow-up surveillance, in which patients were divided into progressive (Pro) and non-progressive (non-Pro) groups, respectively, according to endpoint pathological diagnosis (Table 1). Using the AITongue model to score the PLGC risk for each patient in the follow-up cohort, we found that risk scores from the Pro group were significantly higher than those from the Reg group. The AUC value was 0.71, which showed a significant increase (10.94%, 0.71 vs. 0.64, *p* < 0.01) compared with that derived from the model only including baseline indicators (Figure 5). In addition, we performed a univariate analysis of the TDLs for risk prediction of PLGC (Appendix A). It was found that the TDLs showed limited value for PLGC risk prediction, although they have an enhanced effect on PLGC screening. Therefore, tongue image characteristics are potentially valuable in PLGC risk prediction.

## 4. Discussion

We found that H. pylori infection was weakly correlated with PLGC and non-PLGC, although Hp infection is the most prominent risk factor for GC. Similar results have been found in other studies on the prediction of gastric cancer risk [46,47]. In this study, PLGC was analyzed using symptoms. We found only a small proportion of symptoms correlated with PLGC, and their screening efficiency was not high, which is consistent with the findings of other studies [48,49].

Tongue diagnosis is an important part of the four diagnoses in TCM. In TCM theory, the characteristics of tongue images are quantified into various categories for the diagnosis of diseases [50,51]. In this study, we not only found that helpful feature information for PLGC screening could be extracted through a deep learning model, but also found that these features were related to some categories of tongue diagnosis in traditional Chinese medicine. This suggests that some categories of Chinese tongue diagnosis have interpretable morphological characteristics for PLGC screening and for tongue images of high-risk categories.

Our proposed method has better performance than another study of screening of PLGC. Wang et al. developed a model with non-invasive indicators for PLGC screening based on 290 patients with gastritis, and the AUC was 0.728 (95% CI: 0.651–0.793), whereas the AUC of our method was 0.76 [52].

It is neccessary to adopt cost-effective methods to conduct the large-scale screening for gastric cancer risk in the natural population. Even though gastroscopy and pathological tests are the gold standard for the diagnosis of gastric diseases, they are not suitable for the natural population. The method we developed introduces tongue image information on the basis of conventional and invasive indicators, which improves the accuracy, reduces the difficulty of operation and improve its feasibility for application.

The study has some limitations. The data source was biased compared with the natural population. Due to the need for accurate information on the stage of gastritis, the data for establishing the system all came from patients with gastric disease, which had a certain deviation compared with the natural population. We have developed a smartphone-based app screening system to enhance the application convenience of the AITongue model in the natural population (Appendix A). In further studies, more samples would be collected from natural populations to reduce bias, and larger external validation should be conducted.

## 5. Conclusions

Screening patients with PLGC is important for the prevention and treatment of gastric cancer. In this study, we analyzed the tongue image characteristics associated with PLGC and based on this, constructed a PLGC screening model on a development cohort. It was then externally validated in an independent validation cohort and used to evaluate the capability for risk prediction of PLGC in a follow-up cohort. Our study demonstrates the value of tongue image characteristics in PLGC screening and its potential for risk prediction. 

The screening model constructed in this study could improve the accuracy of PLGC screening. Tongue image characteristics were validated for their value in PLGC screening and risk prediction, which may drive tongue image characteristics as a new risk indicator in the future. By extracting tongue image characteristics through deep learning techniques, this study proposes a new approach for non-invasive PLGC screening and shows the possibility of its use in large-scale applications.

## Figures and Tables

**Figure 1 jpm-13-00271-f001:**
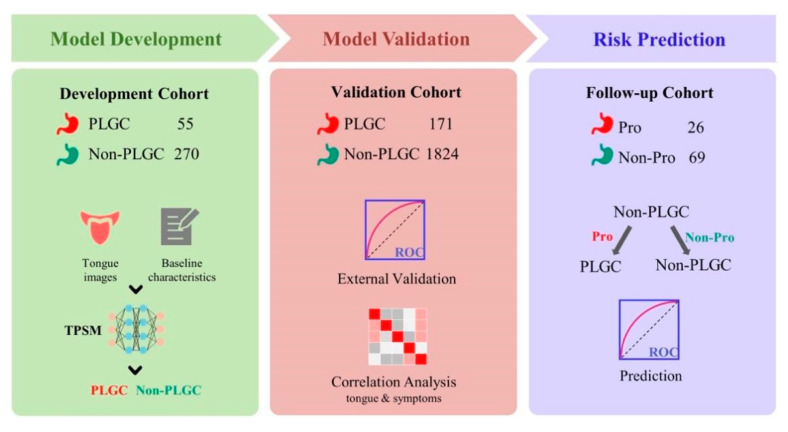
Outline of the workflow for our study.

**Figure 2 jpm-13-00271-f002:**
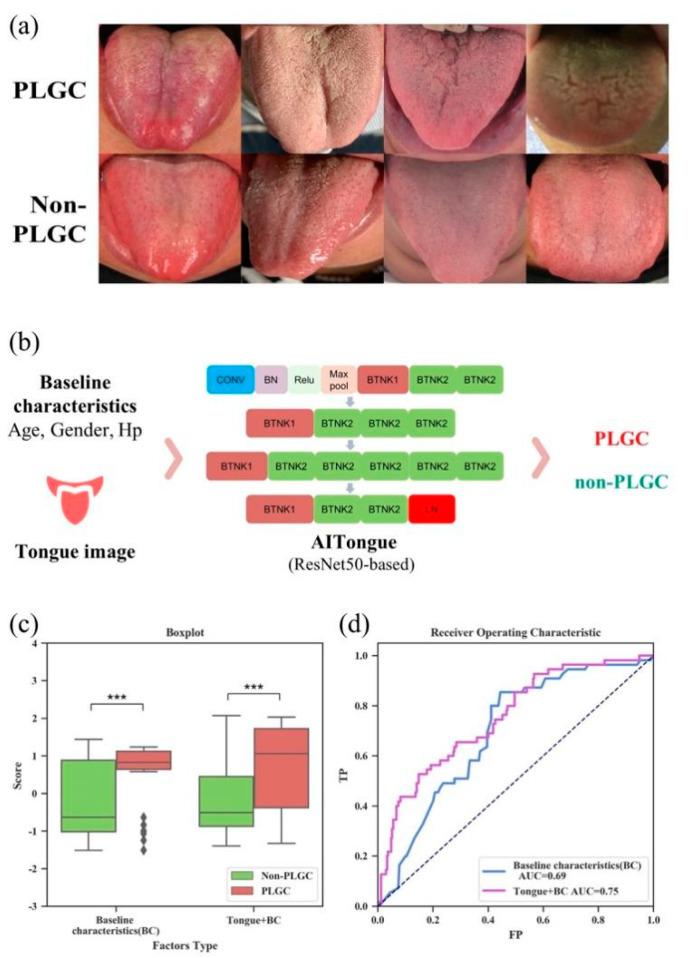
Construction of the AITongue model and results for PLGC screening based on tongue images. (**a**) Example of tongue images of PLGC and non-PLGC patients. (**b**). ResNet50-based deep learning screening model AITongue. (**c**). Boxplot of classification score comparisons with and without the inclusion of tongue images for PLGC screening. (**d**). ROC curves and AUC comparisons for PLGC screening. (***: *p* < 0.001).

**Figure 3 jpm-13-00271-f003:**
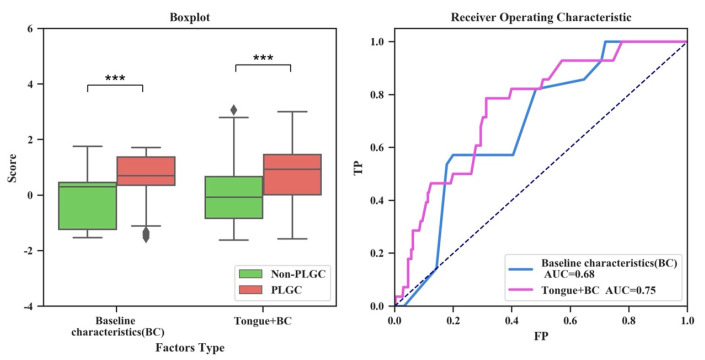
Boxplot (**left**) and ROC curve (**right**) comparisons of screening scores with and without the inclusion of tongue image characteristics for PLGC screening. (***: *p* < 0.001).

**Figure 4 jpm-13-00271-f004:**
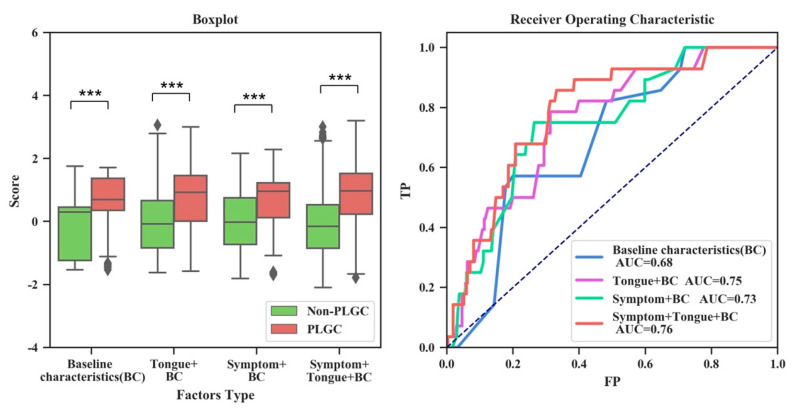
Boxplot (**left**) and ROC curve (**right**) comparisons of the screening scores with and without the inclusion of symptoms factors for PLGC screening. (***: *p* < 0.001).

**Figure 5 jpm-13-00271-f005:**
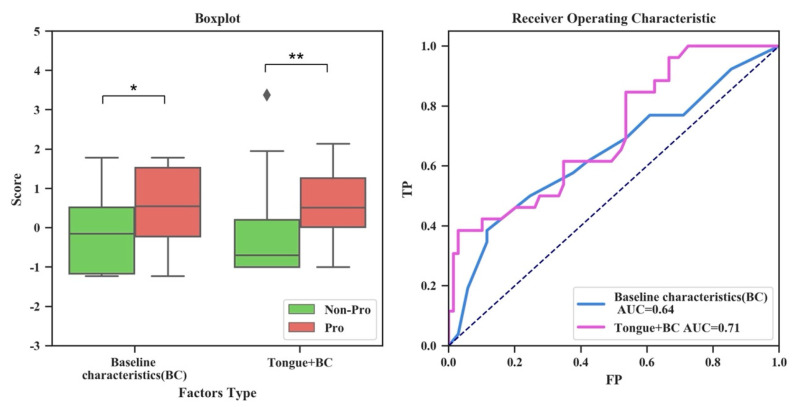
Boxplot and ROC curve comparisons of screening scores with and without the inclusion of tongue image characteristics for risk prediction of PLGC. (**: *p* < 0.01, *: *p* < 0.05).

**Table 1 jpm-13-00271-t001:** Basic information of three cohorts.

Cohorts	Groups	Counts	Age (Mean ± SD)	Gender (Male/Female)
Development cohort	PLGC	55	58.6 ± 10.5	31/24
	non-PLGC	270	48.0 ± 13.6	135/135
Validation cohort	PLGC	171	58.5 ± 9.3	100/71
	non-PLGC	1824	49.9 ± 13.1	685/1139
Follow-up cohort	Pro	26	51.7 ± 8.7	14/12
	non-Pro	69	46.1 ± 11.9	24/45

SD: Squared deviation.

**Table 2 jpm-13-00271-t002:** TDL analysis of high and low-risk tongue images classified by AITongue model.

Characteristics	High Risk (94)		Low Risk (233)		*p* Value
	Counts	Ratio	Counts	Ratio	
Coating (Yellow)	16	0.17	16	0.07	9.6 × 10^−3^
Greasy	28	0.30	34	0.15	2.6 × 10^−3^
Fissured	19	0.20	26	0.11	4.8 × 10^−2^
Coating (Thick)	14	0.15	15	0.06	2.6 × 10^−2^
Dark	12	0.13	7	0.03	1.6 × 10^−3^

*p* value refer to the comparison between high risk and low risk groups by Pearson’s chi squared test.

**Table 3 jpm-13-00271-t003:** Univariate and multivariate analyses of baseline factors and TDLs in PLGC screening.

Variable	Total (N = 1995) No. (%)	Non-PLGC (N = 1824) No. (%)	PLGC (N = 171) No. (%)	*p* Value *	Adjusted OR (95% CI) †	*p* Value #
Baseline Factors						
Age, Years				3.7× 10^−15^		
>50	1067 (0.53)	926 (0.51)	141 (0.82)		4.14 ([2.59,6.61])	2.9 × 10^−9^
≤50	928 (0.47)	898 (0.49)	30 (0.18)			
Gender				1.3 × 10^−7^		
Male	785 (0.39)	685 (0.38)	100 (0.58)		1.47 ([1.00,2.14])	4.8 × 10^−2^
Female	1210 (0.61)	1139 (0.62)	71 (0.42)			
Hp				0.71		
Yes	174 (0.24)	142 (0.24)	32 (0.22)		1.15 ([0.73,1.81])	0.55
No	556 (0.76)	445 (0.76)	111 (0.78)			
Tongue Diagnostic Labels						
Coating				1.9 × 10^−10^		
Yellow	327 (0.16)	269 (0.15)	58 (0.34)		3.66 ([2.38,5.62])	3.4 × 10^−9^
White	1668 (0.84)	1555 (0.85)	113 (0.66)			
Fissure				3.8 × 10^−9^		
Yes	187 (0.09)	149 (0.08)	38 (0.22)		2.26 ([1.37,3.73])	1.4 × 10^−3^
No	1808 (0.91)	1675 (0.92)	133 (0.78)			
Greasy				3.3 × 10^−4^		
Yes	422 (0.21)	367 (0.20)	55 (0.32)		2.23 ([1.44,3.46])	3.3 × 10^−4^
No	1573 (0.79)	1457 (0.80)	116 (0.68)			
Coating				1.3 × 10^−3^		
Thick	239 (0.12)	205 (0.11)	34 (0.20)		2.18 ([1.29,3.67])	3.6 × 10^−3^
Thin	1756 (0.88)	1619 (0.89)	137 (0.80)			
Dark				0.03		
Yes	312 (0.16)	275 (0.15)	37 (0.22)		1.93 ([1.20,3.11])	6.7 × 10^−3^
No	1683 (0.84)	1549 (0.85)	134 (0.78)			

*p* value * refer to the univariate analysis. *p* value # refer to the multivariate analysis (with adjustment for gender and age). † Variables without significance (*p* > 0.05) are not shown.

## Data Availability

Not applicable.

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
