# Peer review of "Construction of Tongue Image-Based Machine Learning Model for Screening Patients with Gastric Precancerous Lesions"

_jpm, 2023, doi:10.3390/jpm13020271_

Round 1
Reviewer 1 Report
The article titled “Construction of tongue image-based machine learning model for screening patients with gastric precancerous lesions” is relatively relevant, however, there are a few concerns that must be addressed and are given as follows: -
1. Objective of the research should be clearly highlighted.
2. Research motivation towards application is insufficient.
3. In practice, the YOLOv5 deep learning model is used for real-time object detection. Why authors used this algorithm? If they used it, then how did they adopt the model architecture for PLGC?
4. In subsection 2.3, the authors said they trained a YOLOv5 model and then in subsection 2.4, the authors said that they utilized the transfer learning ResNet50 model. This is a very serious mistake on the author’s part.
5. It is suggested that the authors need to devise a concrete proposed method section, stating clearly their adopted/proposed model for the PLGC.
6. How do AUC and ROC are computed? Provide a proper equation or computation method for them.
7. The authors need to add a conclusion section that comprehends the overall article.
8. The entire paper's English should be double-checked and improved.
9. There are also numerous typos in the paper that must be revised and corrected.
10. The author should describe more recent relevant research achievements in their manuscript. I propose that the authors introduce some recently proposed ideas as follows.
a. Zhou, H., Liu, Z., Li, T., Chen, Y., Huang, W., & Zhang, Z. (2023). Classification of precancerous lesions based on fusion of multiple hierarchical features. Computer Methods and Programs in Biomedicine, 229, 107301.
b. Yaqoob, M.M.; Nazir, M.; Yousafzai, A.; Khan, M.A.; Shaikh, A.A.; Algarni, A.D.; Elmannai, H. Modified Artificial Bee Colony Based Feature Optimized Federated Learning for Heart Disease Diagnosis in Healthcare. Appl. Sci. 2022, 12, 12080. https://doi.org/10.3390/app122312080
c. Afrash, M.R., Shafiee, M. & Kazemi-Arpanahi, H. Establishing machine learning models to predict the early risk of gastric cancer based on lifestyle factors. BMC Gastroenterol 23, 6 (2023). https://doi.org/10.1186/s12876-022-02626-x.
d. Zhu, X.; Ma, Y.; Guo, D.; Men, J.; Xue, C.; Cao, X.; Zhang, Z. A Framework to Predict Gastric Cancer Based on Tongue Features and Deep Learning. Micromachines 2023, 14, 53. https://doi.org/10.3390/mi14010053.
Reviewer 2 Report
The detection part using Yolov5 is not described in the paper?
The evaluation of the screening model should be done with K cross validation?
line 169 -173: an increase of the accuracy was observed(0.60 to 0.69) on ither hand the sensivity decrease from 0.80 to 0.71 ?
Add the ANOVA test in table 3
